# The Dually Negative Effect of Industrial Polluting Enterprises on China’s Air Pollution: A Provincial Panel Data Analysis Based on Environmental Regulation Theory

**DOI:** 10.3390/ijerph17217814

**Published:** 2020-10-26

**Authors:** Shoujun Lyu, Xingchi Shen, Yujie Bi

**Affiliations:** 1School of International and Public Affairs, Shanghai Jiao Tong University, Shanghai 200030, China; biyujie@sjtu.edu.cn; 2China Institute for Urban Governance, Shanghai Jiao Tong University, Shanghai 200030, China; 3Institute of Healthy Yangtze River Delta, Shanghai Jiao Tong University, Shanghai 200030, China; 4School of Public Policy, University of Maryland, College Park, MD 20742, USA; xcshen@umd.edu

**Keywords:** air pollution, urbanization and industrialization, environmental regulation, polluting enterprises, rigor of regulation

## Abstract

Although the Chinese government has promulgated a series of policies to mitigate air pollution, the air quality in a number of Chinese cities still has the potential to be improved. As the major source of air pollution, enterprises in the industrial and energy sectors are the most difficult to regulate in terms of polluting emissions. This paper aims to investigate what factors influence the intensity of environmental regulations on polluting enterprises based on environmental regulation theory and an empirical test. Firstly, this article builds a theoretic model of optimal regulation supply for local governments in order to examine the relationship between factors influencing the intensity of environmental regulation. Secondly, we use provincial panel data from 2008 to 2015 to test the theoretical hypothesis and use the generalized method of moments (GMM), the two-stage least squares (2SLS) method to address the endogeneity issue. The main finding of the study is that, in regions with a high concentration of polluting enterprises, not only is there more air pollution than in other regions, but the local governments might show partiality towards the polluting enterprises, which could impede the implementation of environmental regulation.

## 1. Introduction

China is the largest developing country and the most populous country in the world, and its rapid process of urbanization and industrialization has brought about heavy pollution. Driven by the rapid growth of industrial production and energy consumption, a number of Chinese cities have reported air pollution levels significantly higher than the health-based standards. Air pollution can spread to surrounding areas and affect a wide range of people. High concentrations of PM2.5 increase the frequency of lung cancer, dyspnea, and heart disease. In the face of intense pressure, the Chinese government has introduced a series of measures and policies, and the policy effects are obvious, which have made the air quality improve in recent years. However, there is still a large gap in air quality between China and developed countries [1]. In Japan, Western Europe, and the U.S., the annual mean population-weighted exposure to PM2.5 is estimated to be below 15 μg/m^3^ on average, while the corresponding index of China as a whole was around 60 μg/m^3^ on average in 2016. The limit annual value is 35 ug/m^3^ according to new ambient air quality standards enacted by the Ministry of Environmental Protection of China in 2012. But in some regions of Eastern China, this index exceeds 100 μg/m^3^, which is 2–7 times the level of developed countries [2].

Sources of air pollution include agricultural production, residential and commercial buildings, power plants, industrial processing, and transportation. Among them, industrial processing and power plants are the main sources. For example, in 2010, the PM2.5 contribution rate of the industrial processing and energy sectors in the Beijing-Tianjin-Hebei region was 51.46% [3]. Air pollution still mainly came from industrial processing in 2015, which contributed more than 50% of the total air pollutant emission.

We choose polluting enterprises in the industrial and energy sectors as the main object of this research for the following reasons. First, the industrial and energy sectors are the major source of air pollution, which exerts a much larger impact on other important social-economic indicators, such as economic growth, employment, consumption, prices, and exports, compared to other air pollution sources. Second, it is more difficult to control air pollution emissions from the industrial and energy sector compared to other pollution sources, because of the problem of information asymmetry between the polluting enterprises and the regulator. Third, there have been relatively certain policy approaches for other pollution sources. For residential and commercial buildings, electrifying buildings with energy efficient technologies provides a pathway to reduce emissions. For the transportation sector, the Chinese government has made efforts to promote a greater market penetration of electrical vehicles. However, it is much harder to find a well-recognized policy approach to fully control emissions from the industrial and energy sectors. Fourth, although the share of industrial added value in gross domestic product (GDP) has continually decreased over the past several years in China, it is still much higher than that of developed countries [4] (See Figure 1). A recent study [1] also shows that the industrial green growth performance was low in China during 2000–2014. Thus, there is still a huge potential to reduce the pollutants emitted by the industrial and energy sectors.

This paper aims to explore the following research questions: What factors influence the intensity of environmental regulation on the industrial and energy sectors? What factors impede the regulating intensity? What is the interacting mechanism behind the regulation structure?

There is abundant literature discussing the impacts of environmental regulation on enterprise performance, but few studies have focused on the influencing factors on the intensity of environmental regulation. This article aims to systemically examine what factors influence the degree of environmental regulation intensity on polluting enterprises in the industrial and energy sectors using environmental regulation theory.

With so many stakeholders involved in smog treatment, the government, playing the important role in the process of regulation, has ineffable difficulties. On the one hand, the Chinese government is under great pressure to reduce smog; on the other hand, the government needs to balance pollution control and economic development. This is a dilemma for air pollution governance, which cannot be characterized as a simple tradeoff between economic development and environmental quality, and behind which are complex relationships between interest groups.

Based on the theory of environmental regulation, this paper provides formal theoretical explanations for the factors that influence environmental regulation of polluting enterprises in the industrial and energy sectors, and tests them empirically. We find a dually significant negative effect of polluting enterprises on China’s air pollution control. In regions with a high concentration of polluting enterprises, not only is there more air pollution than in other regions, but the local governments might show partiality towards the polluting enterprises, and that could impede the implementation of environmental regulation. Finally, it proposes several solutions.

## 2. Literature Review: Environmental Regulation and Air Pollution Governance

There are many practices of environmental regulation theory in solving environmental problems, which can provide insights for air pollution treatment in China. In order to control air pollution, the government must play a leading role as the regulator and collaborate with different social forces to seek the equilibrium of social governance. As a result, it is theoretically and practically significant to study the solutions of environmental problems based on environmental regulation theory.

An overview of studies on environmental regulation shows that coverage of this topic has gradually become heated in recent years. Most research has focused on the impact of environmental regulation on industries or polluting enterprises. In the traditional opinion, an increase in environmental regulation will lead to an increase in enterprises’ pollution treatment costs, thereby weakening enterprises’ competitiveness. However, Porter put forward the famous “Porter Hypothesis” [5], contradicting the traditional point of view. Porter holds the idea that appropriate environmental regulation can motivate enterprises to innovate, resulting in an increase in the efficiency of production. Since then, many scholars have carried out studies on the “Porter Hypothesis”, with some being in favor of the hypothesis, and others questioning it [6,7,8,9,10,11,12,13]. Among these studies, a consistent conclusion is that in the short term, the improvement of environmental regulation will increase enterprises’ costs, thereby reducing their profits, which is not conducive to economic growth.

As for research about air pollution control, although air pollution governance is an important issue in the field of environmental regulation, few scholars have researched it from the perspective of environmental regulation theory. In reviewing the literature about air pollution control, most research focuses on the composition, source analysis, and pathogenic mechanisms of air pollution. Additionally, little research investigates influencing factors on the rigor of environmental regulation. This work aims to remedy the defects stated above.

## 3. A Theoretical Exploration

This section aims to provide a theoretical explanation about what factors can influence the rigor of regulation of polluting enterprises’ airborne emissions based on environmental regulation theory.

Firstly, we can provide a theoretical structure of all of the units related to Chinese local governments’ environmental regulation based on the analysis framework of Stigler and Pletzman’s government regulation model [14,15]. From an empirical perspective, the government’s environmental regulation comes from the demands of the various interest groups concerned with environmental regulation. In the 1970s, the Chicago school, represented by Stigler, first introduced the standard supply–demand analysis method into the research field of government regulation, proposing the theory of interest group regulation. They argued that regulation was derived from the influence exerted by industrial interest groups on public regulators to realize their interests [14]. Since the government has coercive power, industries will seek rent from the government to advance their interests; that is, the industry groups will capture politicians by using material benefits in order to gain government support. Peltzman expanded this theory, including consumer interest groups into the supply–demand model of regulation [14]. Interest groups of environmental regulation can be mainly divided into three types: regulators (government), polluting enterprises, and consumers (the public).

China’s government can be divided into two levels: central government and local government. Relations between central and local governments can be briefly summarized as high decentralization in economic affairs, but high centralization in political affairs [16]. The central government gives orders or instructions to the local government and holds the power of evaluation, appointment, and promotion of local officials. As stakeholders in environmental regulation, polluting enterprises and consumers will constantly put pressure on the government to exert influence on the decision-making of regulation. In addition, the environmental regulation policies of the local government have direct effects on polluting enterprises. Based on the above setting, Figure 2 presents the structure of units related to the local government’s environmental regulation.

Second, the article defines the objectives of the local government’s environmental regulation supply. This has previously been researched from two perspectives: the public interest paradigm and the interest group paradigm [17,18,19]. By summarizing these studies, this article argues that the aim of the local government’s environmental regulation supply is to meet the needs of various interest groups, including itself, as much as possible in order to gain the greatest political support under the binding framework of the central government’s evaluation standard and orders. On this premise, we build the optimal decision-making model of China’s local government environmental regulation, in order to explain which factors determine the level of the environmental regulation intensity of local government. We set the objective function of the regulatory organizations as:(1)$$W=W\left(G_{1},G_{2},G_{3}\right)=L_{1}\cdot G_{1}+L_{2}\cdot G_{2}+L_{3}G_{3}$$

*G*_1_, *G*_2_, and *G*_3_ respectively represent the degree of satisfaction of the public, polluting enterprises, and the local government on environmental regulation, and *L*_1_, *L*_2_, and *L*_3_ respectively represent the level of importance of the three interest groups to the local government. In addition, we set:(2)Gi=Ni•Ui (i=1,2,3)

*N_i_* represents the number of group members, and *U_i_* represents the utility function of the individuals in the group. *R* represents the environmental regulation supply intensity, which is the key variable; *q* is the environmental quality, *π* is the profits of polluting enterprises, *c* is the cost of pollution treatment of polluting enterprises, and *r* is the government taxes.

As for the public, its concern is the quality of the environment. As environmental regulation supply increases, environmental quality will get better, so *dq/dR* > 0, but there will also be a marginal decrease in environmental quality, so *d*^2^*q/d*^2^*R* < 0. *U*_1_, the public’s degree of satisfaction, is closely related to *q*, the environmental quality, so *U*_1_ = *U*_1_
*(q)* = *U*_1_*(R)*, and *dU*_1_/*dR* > 0, *d*^2^*U*_1_/*d*^2^*R* < 0. We therefore use environmental quality as a proxy for the public’s degree of satisfaction, and set *U*_1_ = *q (R)* = *a* + *b*·*ln R*, in which *a* is the basic environmental value and *b* is the coefficient of environmental regulation efficiency.

For polluting enterprises, their utility *U*_2_ is closely related to the profit *π*, and in the short term, the increase of environmental regulation supply *R* will lead to an increase in the cost of pollution treatment *c*, thus reducing the profit *π*. Therefore, *U*_2_ = *U*_2_ (*π*) = *U*_2_
*(R)*, and *dc/dR* > 0, *d**π/dR* < 0, *dU*_2_/*dR* < 0. The marginal cost of pollution treatment will increase with the increase in *R*, so *d*^2^*c/d*^2^*R* > 0, *d*^2^*U*_2_/*d*^2^*R* > 0. We therefore use profit to represent the utility that the polluting enterprises feel, obtaining *U*_2_ = *π(R)* = *P* − *c(R)* = *P* − *εR*^2^, in which *P* is the price of products and is assumed constant, and ε is the coefficient of pollution treatment cost.

For local governments, one of their major demands is to pursue the maximization of fiscal levy [19]. The amount of tax earned by the government is closely related to enterprise profits. As environmental regulation increases, the profit of polluting enterprises will decrease, and the government taxes will decrease accordingly. Therefore, *U*_3_ = *U*_3_*(r)* = *U*_3_*(R)*, *dr/dπ* > 0, *dr/dR* < 0, *dU*_3_/*dR* < 0. We therefore set *r* = *r*_0_ + *k*·*π*, where *r*_0_ represents tax revenue other than that from polluting enterprises and is assumed constant, and *k* is the tax coefficient. We therefore use tax to directly represent the utility that the government perceives, setting *U*_3_ = *r(R)* = *r*_0_ + *k*·(*P* − *εR*^2^).

Therefore, the regulators’ maximum target is:(3)maxW=W(G1,G2,G3)=L1·N1·(a+b·lnR)+ L2·N2·(P−εR2)+L3·N3·[r0+k·(P−εR2) ]

Taking the derivative of W on R and make the first order condition 0, we can get:(4)$$\frac{\partial W}{\partial R}=\frac{bL_{1}N_{1}}{R}-2L_{2}N_{2}\varepsilon R-2L_{3}N_{3}k\varepsilon R=0$$

That is:(5)$$R^{*}=\sqrt{\frac{bL_{1}N_{1}}{2L_{2}N_{2}\varepsilon +2L_{3}N_{3}k\varepsilon }}$$

The second derivative is:(6)$$\frac{\partial^{2}W}{\partial R^{2}}=-\frac{bL_{1}N_{1}}{R^{2}}-2L_{2}N_{2}\varepsilon -2L_{3}N_{3}k\varepsilon <0$$

The second derivative is less than zero, so there is a maximum for *W* when *R = R^*^*, which means that *R^*^* is the optimal intensity of environmental regulation supplied. In the short term, assuming that environmental regulation efficiency coefficient *b*, tax coefficient *k*, and coefficient of pollution treatment cost *ε* are constant, it can be found that the intensity of local governments’ environmental regulation is related to the number of members in the interest groups and their importance to the regulators.

It is assumed that one of the major obstacles for air pollution governance in China is that, although *N*_1_ is much greater than *N*_2_ and *N*_3_, *L*_1_ is much less than *L*_2_ and *L*_3_. In other words, the importance of polluting enterprises and the local government is much bigger than the importance of the public. As a result, it is always difficult for *R* (environmental regulation density) to reach a high level. Detailed explanations are as follows:

Firstly, *L*_3_ (the importance of the local government for itself) is obviously big. As the economic man, the government also has its interest demands, and the coercive power of regulation is in its hands, so it will try to meet its own demands.

Secondly, *L*_1_ (the importance of the public for the local government) is comparatively small, which is related to China’s present political ecology. The Chinese government does not need to pursue the maximization of votes [17]. In addition, although members of the public are large in number, they are sparsely distributed, and the public’s political participation consciousness is low in China [20,21]. Moreover, NGOs, as representatives of public power, are not mature and are relatively weak in strength [22]. Therefore, public pressure on the government is relatively limited.

In addition, *L*_2_ (the importance of polluting industries for the local government) is much greater due to various factors. On the one hand, in the central–local relationship of “fiscal decentralization and political centralization”, local officials are constrained by central assessment and command. Local officials are strongly motivated to seek promotion. Thus, under the central government’s requirement to focus on economic growth, local officials will try as hard as they can to promote the development of the economy by using all of the political and economic resources in their regions [23,24,25,26]. Although in recent years the central government has continuously improved the evaluation system and increased the attention placed on environmental protection, economic growth is still a very important appraisal indicator, especially when China is faced with more pressure due to economic downturn. Emissions are mainly produced by industries that make a large contribution to regional GDP, so they are quite important for the government. In addition, these industries provide a large number of employment opportunities, and the employment rate is very important to regional stability, which is an important index of the central government’s evaluation. Moreover, among these polluting enterprises, there are many state-owned ones, which carry many political functions required by the government. As a result, state-owned enterprises may exert influence on the local government through industry association, banks, or other channels [27]. To summarize, the polluting enterprises are quite important to local governments.

Moreover, the environmental regulation efficiency coefficient *b*, tax coefficient *k*, and the cost of the pollution treatment coefficient ε also deserve attention. By improving *b* and reducing *ε* and *k*, *R* will increase, which is helpful for the improvement of air quality. How to change these coefficients may also be a breakthrough in the attempt to solve the problem of air pollution.

## 4. Empirical Test

In this section, this study aims to use some variables to represent *R, L_1_, L_2_, N_2_/N_1_, ε,* and *b* in order to test the above theoretical assumptions, and we prove that polluting enterprises exert a significant dually negative effect on air pollution control in China.

### 4.1. Main Analysis Units

Before the econometric method and variables are determined, it is first necessary to clarify which polluting enterprises are most responsible for air pollution in China. The direct sources of air pollution are the direct emissions of primary particulate matters and the secondary particles converted from gaseous sulfur dioxide and nitrogen oxides. By consulting the Environmental Statistics Yearbook of China from 2011 to 2015, we can find that six high-pollution industries are the main sources of airborne pollutants, which are: oil processing, coking, and nuclear fuel processing; chemical raw materials and chemical products manufacturing; non-metallic mineral production; ferrous metal smelting and rolling processing; non-ferrous metal smelting and rolling processing; electricity and heat production and supply. Emissions of these six major industries accounted for almost 90% of all industries combined (as shown in Table 1).

We further regress the provincial share of the six high-pollution industries (the contribution rate of the six high-pollution industries to the regional GDP) on different kinds of air quality indexes and find that there is a significant positive correlation between the regional share of polluting enterprises and local air pollution. All of the statistical results are presented in Table 2.

Additionally, under the existing government regulation, the six major industries have all paid a substantial amount to treat emissions, and from 2010 to 2014, the annual operating expenses of emission treatment facilities of the six industries accounted for about 90% of all industries combined (as shown in Table 3).

Thus, we can conclude that the six high-pollution industries are the major industrial sources of air pollution in China. In the face of the government’s environmental regulation, these six high-pollution industries are greatly affected, and other industries are under little influence. Thus, we can take the six major industries as the main research objects.

### 4.2. Econometric Model

This article builds an econometric model using a dynamic panel data model in order to simulate the dynamic behaviors of the local government, which are influenced by its previous behaviors.
y_it_ = α + y_it−1_ + β_1_·X_it_ + ε_it_(7)
ε_it_ = u_i_ + w_it_(8)

The dependent variable y_it_ is the intensity of emission regulation in province i and in period t. Vector X_it_ consists of the explanatory variables of influencing factors; α is the intercept term, ε_it_ is the error term consisting of a time-invariant province-specific random effect u_i_, and an idiosyncratic error w_it_. Assume that: ∀. i ≠ j, t ≠ s
E[u_i_u_j_] = 0, *u_i_*~N(0, *σ**_u_*^2^)(9)
E[*w_it_ w_is_*] = 0, *w_it_* ~N(0, *σ**_w_*^2^)(10)

### 4.3. Data

This work uses panel data from 31 Chinese provinces from 2008 to 2015 to test the econometric model. We do not include data after 2015 because the Environmental Statistics Yearbook of China has not published industrial gas airborne emissions data since 2016.

#### 4.3.1. Dependent Variable

The dependent variable is the intensity of emission regulation, which is represented by the quantity of the pollutant emissions (respectively represented by sulfur dioxide, nitrogen oxide, and soot) per total industrial output value of the six high-pollution industries [28]. Due to the difficulty of obtaining data, we use the emissions of all industries combined as a proxy for the emissions of the six industries. Since the gas emissions in the six industrial sectors all account for over 85% of all industry, the margin of error is within acceptable range. The data were obtained from the statistical yearbooks of 31 provincial units.

#### 4.3.2. Core Independent Variables

Based on the above theoretical derivation, we find several core independent variables as follows:

The importance of polluting industries for the local government, which we use two variables to represent. One is the contribution rate of the six high-pollution industries to the regional economy (economic contribution rate = the proportion of industrial output value of the six industries in all industries combined (the proportion of industrial added value in regional GDP). The other is the proportion of the number of employees in the six high-pollution industries in the regional population. The data were obtained from the statistical yearbooks of 31 provinces.

The importance of the public for the local government, for which we introduce the two variables, the number of environmental petitions and the level of education, to represent. On one hand, the more petitioning people there are and the greater pressure the public puts on the government, the more attention the local government will pay to the public. On the other hand, the higher the citizens’ level of education is, the more likely they will focus on environmental issues, and the more pressure they will put on the government to address them. The variable of environmental petition will be represented by the number of petitioning people per ten thousand regional people, and education level will be represented by the number of students in common colleges and universities out of every ten thousand people in the region. This data came from the Environmental Yearbook of China and the Environmental Statistics Yearbook of China.

#### 4.3.3. Other Control Variables

Fiscal burden of the local government. The difference in governments’ financial burdens that indicates that the capacity to regulate varies widely, affecting regulation intensity. This paper uses the deficit dependency ratio of the local government, the proportion of deficit to fiscal expenditure, to represent this variable. The data were obtained from the website of the National Bureau of Statistics of China.

The coefficient of regulation on pollution treatment cost. We use the number of patents of enterprises above designated size per capita in a province to represent this coefficient, because the enterprise pollution cost is associated with pollution treatment technology, and the technical level is directly related to the number of patents held by the enterprise. The data were obtained from the website of the National Bureau of Statistics of China.

In addition, this work also introduces the regional air pollution level and GDP per capita as control variables. The air pollution level is represented, respectively, by the total emissions per capita of sulfur dioxide, nitrogen oxide, and soot in each province. The data were obtained from the website of the National Bureau of Statistics of China.

#### 4.3.4. Variable Description

The descriptive statistics of the variables studied are shown in Table 4.

In order to understand the pollution situation and polluting enterprises in China directly, we analyze the spatial distribution characteristics of PM2.5 and the influence of polluting industries for the local government.

Figure 3 shows the spatial distribution of PM2.5 index in China during 2013–2018. It can be seen from the figure that the provinces with the largest number of PM2.5 are mainly distributed in the north and central part of China. The PM2.5 in most provinces exceeds the minimum limit of 35 μg/m^3^.

The spatial distribution of the average annual economic contribution rate and the proportion of the employees in population of six polluting industries are shown separately in Figure 4 and Figure 5. A higher economic contribution rate means that local economic development is more dependent on polluting companies. Similarly, a higher proportion of employees represents that employment is more dependent on polluting companies. For the north and central local government, the polluting industries are more important than that for the south local government. From the three figures, we can see that the degree of air pollution is correlated with the influence of polluting industries in China.

### 4.4. GMM 2SLS Method to Address Endogeneity Issue

We use the GMM 2SLS method [29] to examine the above theoretical hypothesis because of the existence of endogenous regressors. Besides the lagged dependent variables, other endogenous explanatory variables, such as the contribution of pollution industries to the regional GDP, the proportion of the number of employees in polluting industries in the local population, and the number of patents of enterprises above a designated size, will also bring about endogeneity bias. The regulation intensity (outcome variable) will also exert a reverse causal effect on the enterprises (independent variables). Under the heavy pressure to reduce emissions from the local government, polluting industries will make adjustments, such as moving to another region where the regulation intensity is comparatively lower, reducing the production scale, laying off employees, or investing more in research and development. Therefore, we adopt the difference GMM 2SLS method introduced by Arellano and Bond [29], which utilizes potential lagged variables (predetermined variables) as instrumental variables, together with the generalized method of moments to address the problem of endogeneity.

### 4.5. Results

In order to improve the robustness of this research, this paper used the emission of sulfur dioxide, nitrogen oxide, and soot to calculate the intensity of emission regulation and introduced three models to examine the above theoretical assumptions. The results of regression are shown in Table 5.

The Arellano–Bond test for zero autocorrelation in first-differenced errors showed that there was no existence of autocorrelation of disturbing term in the three models, and the Sargan test of over identifying restrictions showed that all instrumental variables used in the three models were effective.

The regression results revealed that the proportion of employees in polluting industries in the regional population in model1 and model2, and the GDP contribution rate of the polluting enterprises in model3, were all positively related to the emission per production. In other words, they exerted a negative impact on the regulation intensity. This may mainly be due to the government attaching great importance to GDP growth index and the polluting industries capturing regulators. The higher the enterprise’s economic contribution rate is, the more important it is to the local government, and the higher its dominant position in the region, the greater its ability to influence the decisions of the local government and impede the implementation of environmental regulation.

Additionally, the number of patents of enterprises above a designated size per capita and citizens’ education level both had a significant negative correlation with regulation intensity. Improvements in technology will effectively reduce the cost of pollution treatment, which will be conducive to enhancing the intensity of environmental regulation according to the above theoretical derivation. As the citizens’ level of education improves, they will pay more attention to environmental protection issues, and will strongly urge governments to implement environmental regulation, which will heighten the rigor of the regulation.

However, the work has found that there was no significant relationship between regional environmental petition numbers and the intensity of airborne exhaust regulation. The reason for this may be that the number of petitioners cannot effectively represent L_1_ (the importance of the public for the local government), which means that the petition cannot effectively improve the public’s importance to the local government.

To summarize, the test results were basically consistent with the theoretical hypothesis, which verified the above theoretical derivation.

Although our empirical analysis did not include data of the last three years (2016–2018), we believe that our findings of statistical analysis still make sense, because China is faced with more pressure from the economic downturn of recent years, and the six high-pollution industries should remain quite important for the government.

## 5. Discussion and Policy Implications

With the theoretical derivation and an empirical test, we have investigated what factors influence the environmental regulation density on polluting enterprises’ emissions and find an obstacle for China’s air pollution control, which can be summed up as the game equilibrium of the local government’s environmental regulation shows partiality towards polluting enterprises, causing the game equilibrium between competing interest groups to deviate from the maximization of social welfare. Since the original intention of the environmental policy system is to pursue maximum social welfare, there is a divergence between the institutional arrangement and the real game equilibrium, which is a big obstacle to institutional implementation, further showing up as the weakness of law enforcement in the process of PM2.5 governance. This phenomenon can be summed up as being a collective irrational result caused by individual rational behaviors.

Although the Chinese government has introduced many policies to control air pollution and there has been an obvious decrease in PM2.5 concentration level in recent years, the air quality is still not good enough. The Chinese government could experience a bottleneck stage under the current circumstance of economic turndown. The six traditional high-pollution industries play an important role in economic growth and social stability, and have a close connection to the government, which will exert much more pressure on the government’s environmental regulation.

It is not a simple trade-off between economic growth and phasing out polluting industries. The solution to this problem is to seek a mechanism under which the collective rationality can be realized on the premise of meeting the demand of individual rationality. In other words, we should realize the unity of the game equilibrium of local interest groups and the goal of social welfare maximization through some mechanisms. Two types of steps can be taken: those that weaken the dominance of polluting companies, and those that enhance the influence of the public. This paper makes several suggestions to achieve those ends.

Firstly, improvements should be made to the performance evaluation of local governments on the part of the central government. Emphasizing economic development is beneficial, but not when it comes at the expense of social welfare. Further, the central government should force local governments to attach more importance to sustainable development and to pay more heed to public opinion.

Secondly, industrial structures should be adjusted and environmentally friendly enterprises should be supported. The government should provide more support for environmentally friendly enterprises, guiding their rapid development. Moreover, the development of service industries should be promoted in such a way as to adjust the industrial structure, which can help reduce the dominance of polluting enterprises and be conducive to the enhancement of environmental regulation.

Thirdly, the pollution treatment technology of enterprises should be improved. The government and enterprises should increase investment in research to reduce the cost of pollution control, and encourage the adoption of treatment technology by means of law or policy.

Fourthly, supervision should be strengthened and corruption should be punished. The government should increase the severity of punishment for officials’ illegal behavior in order to prevent officials being corrupted by polluting enterprises. The central government should inspect local governments and large state-owned enterprises more rigorously, and focus particularly on environmental protection issues.

Lastly, the influence of the public should be increased. Efforts to improve education should be continued. With greater education, citizens will pay more attention to environmental issues, and their political participation consciousness will improve, thus exerting greater impact on government decision-making. Citizens should also have a right to know about environmental governance guaranteed. Local governments should establish a sound system of environmental information disclosure, taking the initiative to publicize and explain the environmental protection policies to the public. In addition, it would be beneficial to increase and improve the channels of the public’s political participation. More public meetings and hearings should be encouraged before the implementation of environmental policies in order to give a fair hearing to public opinion. The public can interact with the government and enterprises through social media and other channels to obtain information regarding pollution and progress made in addressing environmental problems.

## 6. Conclusions

This study attempts to reveal what factors influence the environmental regulation density on polluting enterprises’ emissions in China based on environmental regulation theory and an empirical test. We find that the greater the polluting enterprises’ influence is, the more partiality the local government will show towards them, thus leading to an imbalance of regulation and impeding increased force of emission regulation. This will exert a dually negative effect on the improvement of air quality; a great number of polluting enterprises discharge a large amount of exhaust gas and at the same time impede the strengthening of the local government’s regulation of gas emissions. More innovative policy approaches should be applied to address this problem in the future.

## Figures and Tables

**Figure 1 ijerph-17-07814-f001:**
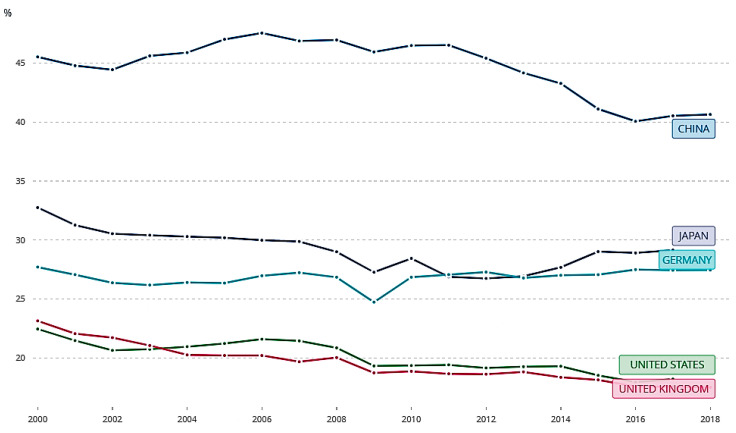
The percentage of industrial (including construction) added value in GDP (data source: World Bank).

**Figure 2 ijerph-17-07814-f002:**
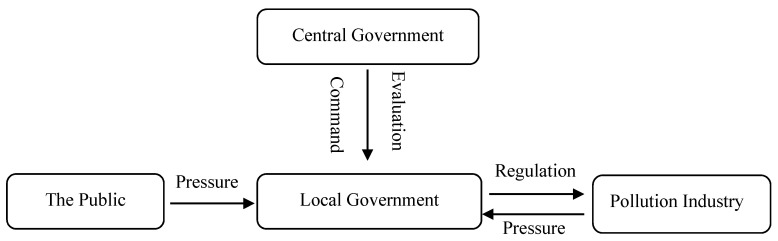
Structure of units related to the local government’s environmental regulation in China.

**Figure 3 ijerph-17-07814-f003:**
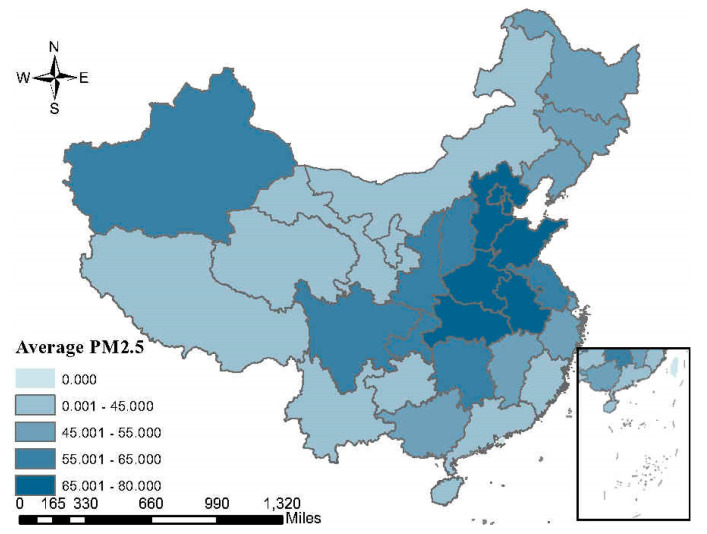
The spatial distribution of average annual PM2.5 index in China. Source: National Meteorological Information Center of China, 2013–2018.

**Figure 4 ijerph-17-07814-f004:**
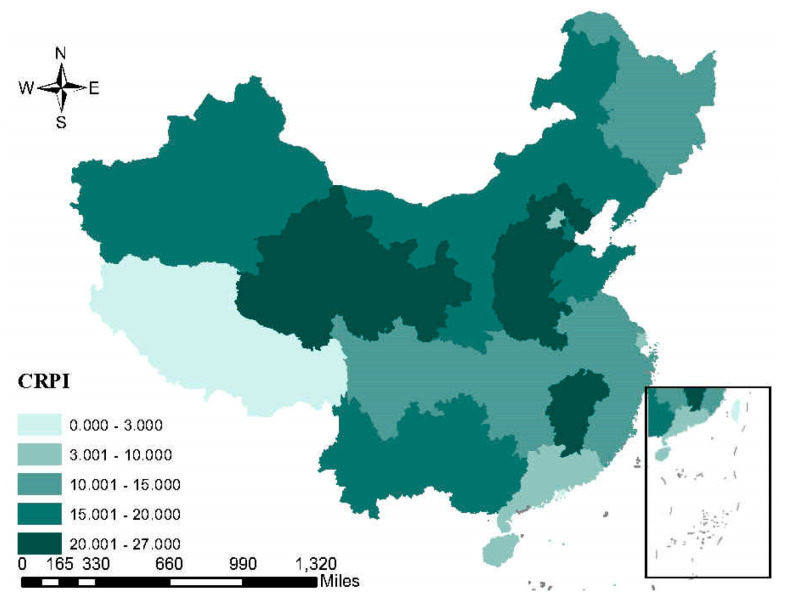
The spatial distribution of average annual economic contribution rate (CRPI) of six polluting industries in China. Source: Calculated by the authors on the basis of the Environmental Statistics Yearbook of China, 2005–2015.

**Figure 5 ijerph-17-07814-f005:**
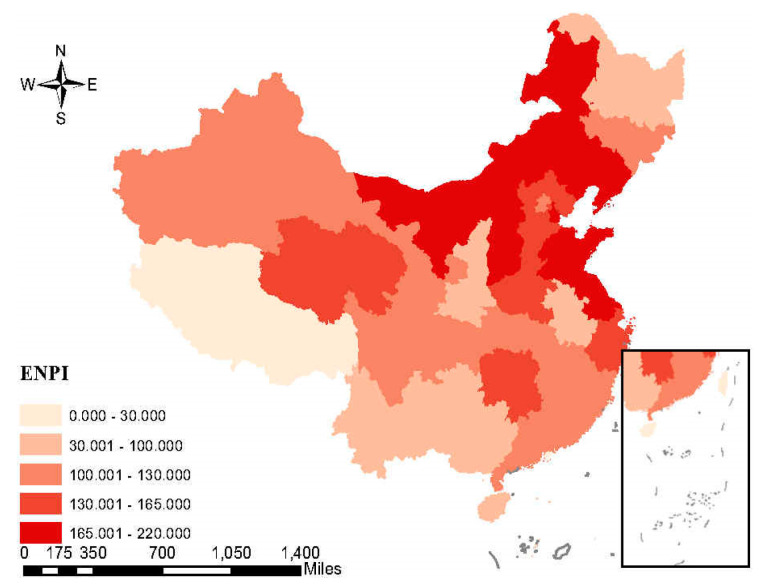
The spatial distribution of average annual proportion of the employees in population (ENIP) of six polluting industries in China. Source: Calculated by the authors on the basis of statistical yearbooks of 31 provinces, 2005–2015.

**Table 1 ijerph-17-07814-t001:** Gas emissions of the six high-pollution industries from 2010 to 2014.

Emission of Pollutants	Year	Proportion of Emissions of the Six High-Pollution Industries Out of All Industries
Emissions of Industrial sulfur dioxide (ton)	2010	87.54%
	2011	88.44%
	2012	87.75%
	2013	87.71%
	2014	87.15%
Emissions of Industrial Nitrogen Oxides (ton)	2011	94.95%
	2012	95.15%
	2013	94.74%
	2014	93.98%
Emissions of Industrial Smoke/ Soot (ton)	2010	84.95%
	2011	82.89%
	2012	82.42%
	2013	84.04%
	2014	87.54%

Data Sources: Environmental Statistics Yearbook of China of 2011, 2012, 2013, 2014, and 2015.

**Table 2 ijerph-17-07814-t002:** The contribution of the six polluting industries to local air pollution (2013–2018).

Regression Model	(1)	(2)	(3)	(4)	(5)
	PM2.5	AQI	SO_2_	NO_2_	CO
The share of six high-pollution industries	119.94 **	153.59 ***	240.05 ***	32.37	3.60 ***
	(49.89)	(54.12)	(37.64)	(23.86)	(0.64)
Constant	43.65 ***	73.13 ***	0.34	32.48 ***	0.72 ***
	(6.95)	(7.53)	(5.24)	(3.32)	(0.089)
Observations	149	149	149	149	149
Number of provinces	31	31	31	31	31

Note: The outcome variables are respectively: annual average PM2.5 concentration (μg/m^3^), AQI (air quality index), annual average SO_2_ concentration (μg/m^3^), annual average NO2 concentration (μg/m^3^), and annual average CO concentration (mg/m^3^) in each province from 2013 to 2018. We obtained the air quality data from the National Meteorological Information Center of China. The independent variable is the contribution rate of the six high-pollution industries to the regional economy (GDP), which is obtained from each province’s statistical year book. Standard errors are in parentheses. Significance level is *** *p* < 0.01, ** *p* < 0.05.

**Table 3 ijerph-17-07814-t003:** Annual operating expenses of waste gas treatment facilities in the six high-pollution industries from 2010 to 2014.

Industry	2014	2013	2012	2011	2010
Sum of all industries (ten thousand Yuan)	17,309,816	14,977,779	14,522,520	15,794,758	10,545,219
Sum of the six main polluting industries (ten thousand Yuan)	16,005,450	13,822,393	13,451,293	14,291,786	8,964,554
Proportion of expenses of the six main polluting industries out of all industries	92.46%	92.29%	92.62%	90.48%	85.01%

Data Sources: Environmental Statistics Yearbook of China of 2011, 2012, 2013, 2014, and 2015.

**Table 4 ijerph-17-07814-t004:** Variable description.

Variable	*n*	Minimum	Maximum	Mean Value	Standard Deviation
Regulation Intensity (SO_2_)	226	3.97	433.48	103.06	84.78
Regulation Intensity (NO)	226	4.83	247.99	64.31	64.31
Regulation Intensity (soot)	226	2.33	196.79	51.87	40.37
Economic Contribution Rate	226	2.38	32.25	14.98	5.68
Employee Proportion	203	24.35	260.14	132.44	51.66
Environmental Petitions	248	0	4.270	0.668	0.507
Education Level	248	74.388	334.81	176.48	55.55
Patents	217	0.033	16.319	1.84	2.70
Deficit Dependency Ratio	248	−211.6	93.6	48.52	26.66
GDP Per Capita	248	9904.22	106,904.9	40,004.88	21,045.85
Pollution (SO_2_)	248	5.53	634.28	185.92	129.67
Pollution (NO)	226	18.29	717.01	187.50	126.46
Pollution (soot)	248	3.42	418.40	106.72	92.65

**Table 5 ijerph-17-07814-t005:** The regression results by the GMM 2SLS method (2008–2015).

	(1)	(2)	(3)
VARIABLES	Regulation Intensity (SO_2_)	Regulation Intensity (NO)	Regulation Intensity (soot)
Regulation intensity (SO_2_)_t−1_	0.483 ***		
	(0.102)		
Regulation intensity (SO_2_)_t−2_	−0.0194		
	(0.0519)		
Regulation intensity (NO)_t−1_		−0.255	
		(0.187)	
Regulation intensity (NO)_t−2_		0.0978	
		(0.0787)	
Regulation intensity (soot)_t−1_			−0.788 ***
			(0.247)
Regulation intensity (soot)_t−2_			−0.660 **
			(0.310)
Number of employees in polluting industries/regional population	0.332 **	0.337 **	−0.0356
	(0.132)	(0.132)	(0.286)
GDP contribution of polluting industries	−266.6	−65.34	491.9 *
	(173.2)	(70.34)	(267.6)
Number of patents of enterprises	−1.632 *	−0.639 *	−0.773 *
	(1.434)	(0.992)	(2.507)
Environmental petitions	−1.665	0.870	3.819
	(2.333)	(1.938)	(3.165)
Education level	−0.601 *	−0.496 *	−1.197 **
	(0.312)	(0.299)	(0.568)
GDP per capita	−0.00152 ***	−0.000831 *	−0.000342
	(0.000456)	(0.000437)	(0.000787)
Fiscal burden of the local government	4.642	0.235	9.894 **
	(2.833)	(0.718)	(3.908)
Pollution (SO2)	−0.307 ***		
	(0.0427)		
Pollution (NO)		0.320 ***	
		(0.0867)	
Pollution (soot)			0.456 ***
			(0.0837)
Constant	269.1 ***	198.8 ***	380.1 ***
	(93.70)	(65.79)	(140.4)
Observations	81	81	81
Number of region	31	31	31

Note: No more than three stage-lagged explanatory variables were used as instrumental variables in all models. Endogenous variables are: the number of employees in polluting industries/regional population, GDP contribution of polluting industries, number of patents of enterprises, pollution (SO2), pollution (NO), and pollution (soot), and from one to three stage-lagged variables were used as instrumental variables. Robust standard errors are in parentheses. Significance level is *** *p* < 0.01, ** *p* < 0.05, * *p* < 0.1.

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
