# Peer review of "The Dually Negative Effect of Industrial Polluting Enterprises on China’s Air Pollution: A Provincial Panel Data Analysis Based on Environmental Regulation Theory"

_ijerph, 2020, doi:10.3390/ijerph17217814_

Round 1
Reviewer 1 Report
Manuscript ID: ijerph-930964
Title: The Dually Negative Effect of Polluting Enterprises on China’s Air Pollution: A Provincial Panel Data Analysis based on Environmental Regulation Theory
Briefly
The authors describe the air pollution problems in China that are producing mostly from industrial and energy sources and they are tried to show the most difficult steps in order to regulate them in terms of Environmental regulation.
As it is, the article has several problems, which should be revise in the next submission
General concern:
- The manuscript has a Word repetition (28 times the word ‘this paper’ is written)
- Abbreviations is missing for (GDP)
- Why the Authors focused only on the air pollution, especially on PM2.5 emission? What are the reasons behind it, please revise this to the introduction and discussion parts? How about other sources of emission, like vehicle emissions.
- Do the industrial activities produce air pollution only? How about water and sediments to soil pollution? How all pollution sources effect on environment and public health? Please revise this issue to the pollution problems and degree of the public satisfaction.
- How about the china’s or international limit values for air pollution? What is the allowed values for the air pollution produce by industrial activities.
- Soot are particulate matter after being emitted and generated, so it is not gas emission.
- What the authors suggest from the results of this study to protect of public health and what action or strategies should be taken to reduce air pollution exposure on human health? Or proposing to better assessment for measuring public/human exposure?
- The use of English language should be improved; there are a few grammatical errors in the manuscript (Word choice, word missing, preposition, etc.). Please proofread the manuscript very carefully to guarantee that the final version is error-free.
Specific concern:
- Title: The chosen title might revised based on the content, since it is reporting the industrial sources air pollution on china’s….
- Abstract: I agree with the abstract text
- Keywords: Please revise them contain some word reputation, and some words are long!
- Introduction: the state of the art of the study and the aims needs to be reorganize in a text under the title of the introduction because other tiles of Literature review, theoretical explanation are confusing the reader to follow the text.
- The title of Materials and Methods is missing in the manuscript,
- Results: the table 5 is only below of the result section, but the other tables need to subscribe in the result section, it is again confusing for the reader to follow the result section.
7) Discussion: this study has not discussion at all; authors need to arrange and formulate their finding and discussing them in a scientific way to proof of their work that has impact on the air pollution study.
8) From conclusions part, it is suggesting to authors to re write the conclusion based on the obtained results, showing the correlation between of the air pollution and public satisfaction health effects (outcomes) in population/local government are studied. However it is difficult to relate Data analysis results to sources and routes of exposure of environmental chemicals for develop effective health risk management strategies but for regulating the way of industrial air pollution production (like using filter technology) , and the effect on personal protective measures might be required.
Author Response
Response to Reviewer #1
We thank the reviewer for constructive comments that greatly help improve the technical quality and the presentation of this manuscript. We have carefully corrected and supplemented the paper as the reviewer suggested. We hope this can improve the quality of this article. The detailed responses and modification to the comments are listed as follows:
- Original Comments:
“The manuscript has a Word repetition (28 times the word ‘this paper’ is written)”
Response: Thank you for your careful reading of our manuscript. In the new version, we have replaced the words.
- Original Comments:
“Abbreviations is missing for (GDP)”
Response: ‘Gross domestic product (GDP) ’ has been supplemented.
- Original Comments:
“Why the Authors focused only on the air pollution, especially on PM2.5 emission? What are the reasons behind it, please revise this to the introduction and discussion parts?”
Response: There are different kinds of pollution sources. Air-pollution is one of the most serious environment problems in China in recent year. This study is focused on investigating air pollution. Air pollution can spread to surrounding areas and affect a wide range of people. In addition, PM2.5 is a representative index to measure the extent of air pollution. High concentrations of PM2.5 increase the frequency of lung cancer, dyspnea and heart disease. Government, the public, and the media all concern this problem in China. Your suggestion is very pertinent. The relevant content has been supplemented in the introduction and discussion.
- Original comments:
“How about other sources of emission, like vehicle emissions.”
Response: The emission from vehicles also contributes to air pollution, but this study is focused on investigating polluting enterprises. Detailed description can be seen in the 2th and 3th paragraph of introduction. “Industrial processing and power plants are the main sources. For example, in 2010 the PM2.5 contribution rate of the industrial processing and energy sectors in the Beijing-Tianjin-Hebei region was 51.46% (Guan and Liu, 2013)….”. Compared with other sources of emission, like vehicle emissions,it is more urgent and difficult to provide useful and efficient regulation on polluting enterprises’ emissions compared to other emission sources. According to your comment, we may consider the traffic pollution in future research.
- Original Comments:
“Do the industrial activities produce air pollution only? How about water and sediments to soil pollution? How all pollution sources effect on environment and public health? Please revise this issue to the pollution problems and degree of the public satisfaction.”
Response: Industrial activities will also produce other pollution. But the proportion of industrial waste water in total waste water is about 20%-30%, according to the China Environmental Statistics Yearbook. Air pollution still mainly comes from industrial processing in 2015, which contributes more than 50% of the total air pollutant emission. We aim to study the environmental regulations, and most environmental regulation direct to polluting enterprises. So air pollution is more representative. (Relevant content can be seen in the first three paragraphs in the introduction section). In addition, many previous studies explored the effect of air pollution and draw conclusions (Wang et al. 2019; Yang et al. 2019).
We give attention to the degree of the public satisfaction, but don't just focus on public satisfaction, but also the degree of satisfaction of the public on environmental regulation, and the degree of satisfaction of the local government on environmental regulation. In section 3 (A Theoretical Exploration), we define that: G1 represents the degree of satisfaction of the public on environmental regulation; G2 represents the degree of satisfaction of polluting enterprises on environmental regulation; G3 represents the degree of satisfaction of the local government on environmental regulation. Then, we set the objective function of the regulatory organizations as:
So, we aims to investigate how public, polluting enterprises, and other factors influence the intensity of environmental regulations on polluting enterprises.
- Original Comments:
“How about the china’s or international limit values for air pollution? What is the allowed values for the air pollution produce by industrial activities.”
Response: The limit annual value is 35μg/m3 according to new ambient air quality standards, enacted by the Ministry of Environmental Protection of China in 2012. Thanks for your reminder. We have supplemented explanations in result part and in the first paragraph of introduction. The new Ambient Air Quality Standards can be found from the following website:
- Original Comments:
“Soot are particulate matter after being emitted and generated, so it is not gas emission.”
Response: Although soot is not gas, it is considered to be one main source of air pollution or PM2.5(Zhang et al. 2013). In addition, in China Environmental Statistics Yearbook, soot is categorized into the part of emission of waste gas.
- Original Comments:
“What the authors suggest from the results of this study to protect of public health and what action or strategies should be taken to reduce air pollution exposure on human health? Or proposing to better assessment for measuring public/human exposure?”
Response: At present, we think government regulation is still the main force to curb air pollution. As shown in Figure 2 in the paper, it is difficult for the public to directly influence polluting enterprises, and they mainly put pressure on the central government to supervise the local government. In the last part (Conclusion and Policy Implications) of the article, we summarize useful conclusions and policy implications for environmental regulation from the perspectives of government, industrial structure, public, and so on. The government and the public can utilize these suggested measures to improve air pollution control and public health.
- Original Comments:
“The use of English language should be improved; there are a few grammatical errors in the manuscript (Word choice, word missing, preposition, etc.). Please proofread the manuscript very carefully to guarantee that the final version is error-free.”
Response: Thanks for your patience and suggestions. Our manuscript has been proofread and edited in the new version.
- Original Comments:
“Specific concern: Title: The chosen title might revise based on the content, since it is reporting the industrial sources air pollution on china’s…”
Response: Thanks for the reviewer for this comment. We have changed our title to “The Dually Negative Effect of Industrial Polluting Enterprises on China’s Air Pollution: A Provincial Panel Data Analysis based on Environmental Regulation Theory”.
- Original Comments:
“Abstract: I agree with the abstract text
Keywords: Please revise them contain some word reputation, and some words are long!”
Response:We have simplified the long words in the Keywords.
- Original Comments:
“Introduction: the state of the art of the study and the aims needs to be reorganize in a text under the title of the introduction because other tiles of Literature review, theoretical explanation are confusing the reader to follow the text.”
Response: We thank the reviewer for this comment. In order to state the aim of our study more clearly in the introduction, we add a new paragraph “This paper aims to explore the following research questions: What factors influence the intensity of environmental regulation on the industrial and energy sectors? What factors impede the regulating intensity? What is the interacting mechanism behind the regulation structure?”
Moreover, in the last paragraph of the introduction section, we add a summary of our findings in this paper, “We find a dually significant negative effect of polluting enterprises on China’s air pollution control. In regions with a high concentration of polluting enterprises, not only is there more air pollution than in other regions, but the local governments show partiality towards the polluting enterprises, impeding any efforts to strengthen the intensity of environmental regulation.”
- Original Comments:
“The title of Materials and Methods is missing in the manuscript.”
Response: Materials are listed in the part 4.3, and methods is in the part 4.4.
- Original Comments:
“Results: the table 5 is only below of the result section, but the other tables need to subscribe in the result section, it is again confusing for the reader to follow the result section.”
Response: We thank the reviewer for this comment. Table 5 is the only table presenting analysis results. Other tables only serve as background information.
- Original Comments:
“Discussion: this study has not discussion at all; authors need to arrange and formulate their finding and discussing them in a scientific way to proof of their work that has impact on the air pollution study.”
Response: Thank the reviewer for this reminder and suggestion, we have added a discussion section in the new version. And in the last part “Conclusion and Policy Implications” of the article, we summarize useful conclusions for environmental regulation from the perspectives of government, industrial structure, public, and so on.
- Original Comments:
“From conclusions part, it is suggesting to authors to re-write the conclusion based on the obtained results, showing the correlation between of the air pollution and public satisfaction health effects (outcomes) in population/local government are studied. However, it is difficult to relate Data analysis results to sources and routes of exposure of environmental chemicals for develop effective health risk management strategies but for regulating the way of industrial air pollution production (like using filter technology) , and the effect on personal protective measures might be required.”
Response: Based on the results, we draw conclusions that there is no significant relationship between public influence and the intensity of environmental regulations on polluting enterprises. Therefore, in the last part, we put forward some advice to increase the influence of the public.
Although our analysis does not have direct connection to the personal health protective measures towards the air pollution, we analyze the environmental pollution problem from the perspective of regulation structure. We introduce several measures to increase the intensity of regulation on polluting enterprises’ emissions in the last section. Following our suggested strategies, the air quality will be improved eventually and people’s health can also be improved.
References cited:
Wang K, Yin H, Chen Y (2019) The effect of environmental regulation on air quality: A study of new ambient air quality standards in China. Journal of Cleaner Production 215:268-279
Yang F, Muhamad JW, Yang Q (2019) Exploring Environmental Health on Weibo: A Textual Analysis of Framing Haze-Related Stories on Chinese Social Media. International Journal of Environmental Research and Public Health 16(237413)
Zhang R, Jing J, Tao J, Hsu SC, Wang G, Cao J, Lee CSL, Zhu L, Chen Z, Zhao Y, Shen Z (2013) Chemical characterization and source apportionment of PM2.5 in Beijing: seasonal perspective. ATMOSPHERIC CHEMISTRY AND PHYSICS 13(14):7053-7074

Reviewer 2 Report
The manuscript 'The Dually Negative Effect of Polluting Enterprises on China’s Air Pollution: A Provincial Panel Data Analysis based on Environmental Regulation Theory' presents important and timely analysis. I enjoyed reading through the draft which flows lucidly. I feel that it should be accepted after addressing the below concerns.
i) As the pollution over most of China started decreasing effectively from 2014, I would suggest the authors to increase the time period from 2008-2015 to more recent years
ii) I would suggest the authors to depict spatially the density of high emission industries and emission of the pollutants on a map for the audience to have a wider view of the situation.
iii) Is there a way to show the intensity of dual-negative impeding factor by region? As far as I understand, it should mimic the distribution of density of high emission industries in a state? Such representation would enhance the quality of the paper.
Author Response
Response to Reviewer #2
We thank the reviewer very much for the comments and suggestions, which guide us to clearly denote the real research content of the work. We have tried our best to improve the manuscript and made some changes in the manuscript.
- Original Comments:
“As the pollution over most of China started decreasing effectively from 2014, I would suggest the authors to increase the time period from 2008-2015 to more recent years”
Response: We thank the reviewer for this comment. It could make our paper more relevant and interesting to including the latest data. However, we do not include data after 2015 because the Environmental Statistics Yearbook of China has not published industrial gas airborne emissions data since 2016. The description of data can be seen in 4.3 Data.
Although our empirical analysis does not include data of last three years (2016-2018), we believe that our findings of statistical analysis still make sense because China is faced with more pressure from the recent years’ economic downturn, and the six high-pollution industries should remain quite important for the government. Thus, our main conclusion should still hold. (Relevant content can be found in the end of section 4.5. Results)
Moreover, using data from 2008 to 2015, we draw meaningful results. The results are basically consistent with the theoretical hypothesis, which verify the above theoretical derivation.
- Original Comments:
“I would suggest the authors to depict spatially the density of high emission industries and emission of the pollutants on a map for the audience to have a wider view of the situation.”
Response: Thank the reviewer for this suggestion. We have supplemented three maps and analysis in the last part 4.3.4 (Variable Description) to improve the paper’s quality.
- Original Comments:
“Is there a way to show the intensity of dual-negative impeding factor by region? As far as I understand, it should mimic the distribution of density of high emission industries in a state? Such representation would enhance the quality of the paper.”
Response: We thank the reviewer for this comment. It could make our study much more interesting by including this heterogeneity analysis. However, in this study, since China does not disclose the data of 657 cities or 2856 county-level administrative units, the cross-sectional units in the statistical model can only be China’s 31 provinces-level administrative units. If we run the regression models by region, the sample size for each region could be too small, which cannot allow us to obtain robust estimation results with statistical meaning. Unfortunately, we are unable to conduct this heterogeneity analysis.
References cited:
Wang K, Yin H, Chen Y (2019) The effect of environmental regulation on air quality: A study of new ambient air quality standards in China. Journal of Cleaner Production 215:268-279
Yang F, Muhamad JW, Yang Q (2019) Exploring Environmental Health on Weibo: A Textual Analysis of Framing Haze-Related Stories on Chinese Social Media. International Journal of Environmental Research and Public Health 16(237413)
Zhang R, Jing J, Tao J, Hsu SC, Wang G, Cao J, Lee CSL, Zhu L, Chen Z, Zhao Y, Shen Z (2013) Chemical characterization and source apportionment of PM2.5 in Beijing: seasonal perspective. ATMOSPHERIC CHEMISTRY AND PHYSICS 13(14):7053-7074

Round 2
Reviewer 1 Report
i have no any further comments for the authors
Author Response
Thank you for reviewing this article.